

# Sa-SNN: spiking attention neural network for image classification

Yongping Dan[1], Zhida Wang[1], Hengyi Li[2] and Jintong Wei[1]

[1] School of Electronic and Information, Zhongyuan University of Technology, Zhengzhou, Henan, China
[2] Research Organization of Science and Technology, Ritsumeikan University, Kusatsu, Japan

## ABSTRACT

Spiking neural networks (SNNs) are known as third generation neural networks due to their energy efficient and low power consumption. SNNs have received a lot of attention due to their biological plausibility. SNNs are closer to the way biological neural systems work by simulating the transmission of information through discrete spiking signals between neurons. Influenced by the great potential shown by the attention mechanism in convolutional neural networks, Therefore, we propose a Spiking Attention Neural Network (Sa-SNN). The network includes a novel Spiking-Efficient Channel Attention (SECA) module that adopts a local cross-channel interaction strategy without dimensionality reduction, which can be achieved by one-dimensional convolution. It is implemented by convolution, which involves a small number of model parameters but provides a significant performance improvement for the network. The design of local inter-channel interactions through adaptive convolutional kernel sizes, rather than global dependencies, allows the network to focus more on the selection of important features, reduces the impact of redundant features, and improves the network's recognition and generalisation capabilities. To investigate the effect of this structure on the network, we conducted a series of experiments. Experimental results show that Sa-SNN can perform image classification tasks more accurately. Our network achieved 99.61%, 99.61%, 94.13%, and 99.63% on the MNIST, Fashion-MNIST, N-MNIST datasets, respectively, and Sa-SNN performed well in terms of accuracy compared with mainstream SNNs.

# INTRODUCTION

In the past few decades, the advancement of artificial neural networks (ANN) has propelled the flourishing development of machine learning algorithms (*Jain, Mao & Mohiuddin, 1996*). Inspired by the biological brain, the construction of artificial neural networks is based on neurons, which receive weighted inputs from neurons in the preceding layer and propagate and accumulate these inputs through continuous activation values, while utilizing differentiable nonlinear activation functions. Neurons are organized hierarchically and stacked in layers to form complex network topologies (*Li et al., 2023a*). The differentiable nature of activation functions enables training through gradient-based

Corresponding author
Yongping Dan, 6100@zut.edu.cn

optimization methods such as backpropagation, wherein network parameters are fine-tuned to match expected outputs. Recent advancements in GPU computing power, coupled with the availability of large annotated datasets, have made training deep networks more feasible, leading to the emergence of the research field known as deep learning (DL), encompassing deep neural networks (DNN) with multiple layers (*Zhang et al., 2016*). DNNs have found successful applications in domains such as image recognition, object detection, speech recognition, and Go playing (*Li et al., 2023b*).

Although inspired by biological neural networks, there are fundamental differences in the processing mechanisms between DNNs and SNNs (*Zhang et al., 2016*). The main difference is how the information is represented and delivered. DNNs represent input as continuous activation values, which are passed and processed to downstream neurons through weighted inputs, while biological neurons communicate through spikes, forming a synapse between each pair of neurons connection. Communication between biological neurons is represented by sparse spike timing, or spike frequency, within a specific time window.

Furthermore, the learning process in biological brains differs from the training process in DNNs. The adjustment of synaptic strength between biological neurons depends on the relative timing of input and output pulses, which is completely different from the gradient-based learning method in DNN (*Li et al., 2021*), which optimizes a single loss function and relies on the network connection between each layer.

The main motivations for using SNNs instead of DNNs are as follows: First, SNNs is closer to the working mechanism of biological neurons and can handle dynamic time series data and sparse data more effectively. Secondly, SNNs performs well in energy efficiency and is particularly suitable for edge computing and IoT devices, which can reduce energy consumption while maintaining efficient computing. Finally, since SNNs utilizes the temporal structure of spikes to encode information, it may have stronger advantages in processing temporal information. Therefore, exploring the application of SNNs in certain tasks has important research value.

These observations have driven the development of the third generation of artificial neural networks—spiking neural networks (SNNs) (*Ghosh-Dastidar & Adeli, 2009*; *Tavanaei et al., 2019*). SNNs are designed to closely mimic the operation of the human brain, as communication between neurons occurs through spikes, and weighted connections between neuron pairs can be adjusted through some form of spike-timing-dependent plasticity (STDP) (*Kheradpisheh et al., 2018*). Compared to DNNs (*Zhang et al., 2016*), SNNs offer several advantages: firstly, research indicates that SNNs achieve computational capabilities at least comparable to ANNs while requiring fewer computational resources; secondly, spikes are infrequent in time, and communication is event-driven (*Lin et al., 2022*), resulting in reduced power consumption; finally, SNNs can capture temporal features of data, with spike timing playing a crucial role in various input encoding strategies. Leveraging these advantages, SNNs have been widely applied in fields such as visual processing, speech recognition, and medical diagnosis. In recent years, deep spiking neural networks (DSNNs) (*Li & Meng, 2023*; *Vijayan & Diwakar, 2022*; *DAngelo*
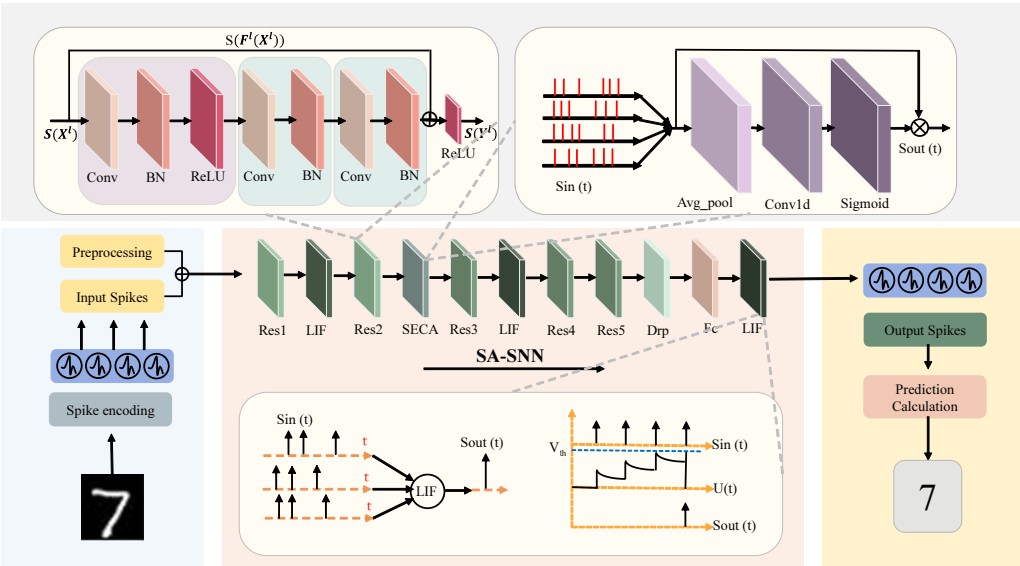

**Figure 1** **SA-SNN network overall architecture design diagram.** The figure integrates the principle diagram of our proposed SECA attention mechanism, the improved residual structure diagram and the working principle diagram of the LIF neurons used.

*& De Zeeuw, 2009*), which combine the multi-layer structure of ANNs with biologically inspired spiking mechanisms, have become a focal point of research.

Traditional neural network models treat all input data as equally important, whereas attention mechanisms assign different weights to different parts of the input data to more accurately focus on important information (*Ghosh-Dastidar & Adeli, 2009*; *Tavanaei et al., 2019*). Specifically, attention mechanisms divide input data into query vectors and key-value pairs, where query vectors represent the target to be focused on, and key-value pairs represent different parts of the input data (*Wang et al., 2020*). By computing similarities, weight values for different parts can be obtained to more precisely focus on important information. Attention mechanisms weight different parts of input data to more accurately focus on important information and have been widely applied in fields such as natural language processing, computer vision, and speech recognition, becoming a crucial technology in the field of deep learning.

The SA-SNN model proposed in this article first converts the image information into a spike signal through spike encoding, and then the spike signal is extracted through a multi-layer spiking convolution network. We introduce a channel attention module, which enhances the response to important features by adaptively assigning channel weights. Specifically, the channel attention module generates attention weights by calculating the global average pooling and maximum pooling features of each channel, and applies them to the feature map to improve the expressive ability of the model. Finally, the extracted features are classified by an spike classifier. The overall framework diagram of this design is shown in Fig. 1. Our main contributions are as follows:

1. Proposed SA-SNN model: We design an efficient spiking neural network incorporating a channel attention mechanism, which can significantly improve image classification performance while maintaining high computational efficiency.

2. Design of efficient channel attention module: By introducing the channel attention mechanism, we adaptively adjust the channel weights and enhance the feature extraction effect.

3. Design of deep residual block: We optimize and deepen the structure of the existing residual network so that the network can extract important information more precisely.

In this paper, we introduce the important developments in SNNs and attention mechanisms in recent years in related work. In methods, we will provide a detailed overview of the implementation methods of Sa-SNN. Specific implementation details will be emphasized in experimental evaluation. Finally, the conclusions will summarize the entire paper.

## RELATED WORK

The human brain relies on neurons to construct intricate neural systems for information processing. Inspired by the brain, ANNs have found widespread applications in the field of artificial intelligence (AI) (*Ghosh-Dastidar & Adeli, 2009*). Despite the remarkable achievements of ANNs in various tasks, there still exists a significant gap between their operational principles and the mechanisms of information processing in the brain. Since the early 1950s, *Hodgkin & Huxley (1952)* began to investigate the electrochemical properties of neurons and described their spiking behavior through equations (*Hodgkin & Huxley, 1952; Li & Meng, 2023*). By the late 1980s, biologists discovered synchronous oscillations in the cat visual cortex, sparking widespread interest in neuroscience. With a growing understanding of brain information processing mechanisms, spike neural networks (SNNs) (*Izhikevich, 2003*), inspired by biological neurons, emerged. SNNs operate in a manner closely resembling the processing of sensory information in the human brain, hence earning them the title of the third generation of neural networks (*Tavanaei et al., 2019*).

Over the past 40 years, biologists studying brain mechanisms have dedicated significant efforts to developing universal SNNs learning and training models (*Moraitis, Sebastian & Eleftheriou, 2018*). The learning and adjustment of synaptic weights in SNNs depend on synaptic dynamics. Researchers found that the sustained response of neurons to synaptic activity manifests as changes in synaptic strength, leading to the proposal of spike-time-dependent plasticity (STDP) (*Kheradpisheh et al., 2018*), recognized as a primary driver of learning and memory in the biological brain. STDP, based on spike-controlled biological mechanisms, is compatible with spike neuron models. In 2000, Rossum et al. introduced an unsupervised Hebbian mechanism based on Hebbian rules, significantly enhancing the stability of STDP learning (*Kheradpisheh et al., 2018*). *Song, Miller & Abbott (2000)* simulated the brain's neural systems using Hebbian-based SNNs in 2000 and 2001. Although the above training mechanisms yield performance comparable to artificial neural networks in shallow networks, their performance in deep SNNs remains unsatisfactory due to SNNs' inability to be trained *via* error backpropagation (BP) methods.

Consequently, approximate error BP algorithms have been proposed in recent years, demonstrating excellent performance (*Wu & Er, 2000*; *Kosko, 1989*). Converting ANNs into SNNs is one of the most effective methods for achieving deep SNNs, although this approach may lead to a decline in network performance (*Bu et al., 2023*; *Ho & Chang, 2021*). Therefore, the development of efficient, universal SNNs algorithms for unstructured perceptual information such as image processing (IP) is crucial for achieving neuromorphic intelligence. Compared to artificial neural networks, SNNs deployed on neuromorphic chips exhibit greater information processing capabilities (*Rathi et al., 2023*). SNNs operate as event-driven and sparsely triggered networks, with computation and energy consumption primarily concentrated at the moments when neurons emit spikes (*Lin et al., 2022*; *Li et al., 2023a*). Unlike artificial neural networks, SNNs effectively save energy consumption for neurons in inactive states. Furthermore, SNNs can be simulated on specialized neuromorphic chips such as IBM's TrueNorth chip (*Akopyan et al., 2015*) and the SpiNNaker superbrain machine (*Painkras et al., 2013*). SNNs continue to evolve rapidly, with their potential applications on mobile devices becoming increasingly evident. SNNs have yielded fruitful research results in the IP domain, spanning multiple fields including informatics, medicine, and socioeconomics. ANNs and SNNs exhibit significant differences in network composition and information processing methods. Unlike artificial neural networks, SNNs lack a comprehensive theoretical framework and mature training methods. Moreover, research on neuromorphic chips is still in its infancy, limiting the depth of SNNs and their applications in complex tasks. This is also the primary reason why SNNs are currently mainly used for simple IP.

# METHODS

Deep residual networks (ResNet) stand as a specialized iteration of convolutional neural networks (CNNs) developed to address the issue of diminishing performance with increasing network depth. These networks have achieved broad adoption due to their capacity to mitigate the degradation problem inherent in training very deep networks (*Fang et al., 2021*).

In the forthcoming section, we will describe the spike coding method we use in 'Spike encoding'. We outline the utilization of spiking neurons, as detailed in 'Spiking neuron model'. Subsequently, in 'Spiking Residual Neural Networks', we introduce the SECA (Spiking-efficient Channel Attention) module, which is meticulously designed to facilitate the acquisition of channel attention. Lastly, the assembly of the spiking residual attention neural network is presented in 'SECA Module for Channel Attention'. This amalgamation of components forms the foundation of the proposed model's architecture, geared towards enhancing the representation and attention capabilities of the network.

## Spike encoding

When the retina converts photons into spikes, we see light. When volatile molecules are converted into spikes, what we smell is odour. When nerve endings convert tactile pressure into stimulation, what we feel is touch. The brain processes current spikes conducted by neurons. If our goal is to build a SNN, then it makes sense to use spikes at the input.

Although non-spike inputs are common, the advantages of encoding data come from the three "S": Spikes, Sparsity, and Static suppression.

Spikes: Biological neurons process and communicate *via* spikes, which are electrical spikes of approximately 100 millivolts. Many computational models of neurons reduce voltage bursts to discrete single-bit events: "1" or "0". In hardware, this is much simpler than high-precision value representation.

Sparsity: neurons spend most of their time in a resting state, with most of their activation states being zero at any given moment. Sparse vectors/tensors (with a large number of zeros) are not only inexpensive to store, but also need to be computed by multiplying the sparse activations with the synaptic weights. If most values are multiplied by "0", there is no need to read a large number of network parameters from memory, which makes neuromorphic hardware very efficient.

Static suppression: Perceptual modules process information only when there is new information to process. Every pixel in the image responds to changes in brightness, so large portions of the image are occluded. Traditional signal processing requires all channels, pixels to follow a global sampling, shutter rate, which reduces how often perception occurs. Event-driven processing now only improves sparsity and power efficiency by blocking immutable input, but it can generally achieve faster processing.

SNNs can utilise time-varying data. However, in this study, MNIST is not a time-varying dataset, and we use a frequency coding approach. Frequency coding is a coding strategy in SNNs that represents the intensity of the input signal through the frequency of spike firing. In this encoding method, the intensity of the input signal is mapped to the frequency of neuron spike firing. In other words, the stronger the signal, the higher the frequency of spike emission; the weaker the signal, the lower the frequency of spike emission, or even no spike emission, as shown in Fig. 2. The same training sample is repeatedly passed to the network at each time step. This is like converting MNIST into a static, unchanging video. Each element can take a high precision value normalized between 0 and 1.

## Spiking neuron model

The simplest and most commonly used mathematical model of pulsatile neuron potential dynamics is the leaky integrate-and-fire (LIF) model, which works in a process similar to the charging, leakage, and discharging processes of biological neurons. A more accurate description of the biological neuronal dynamics model is the Hodgkin–Huxley model (*Nelson & Rinzel, 1995*), but the differential equations of this model are complex and difficult to understand intuitively. Although the Hodgkin–Huxley model more accurately describes the process of potential kinetic changes in biological neurons, it is believed that this model does not fit the data as well as the LIF model (*Fang et al., 2021*). In conclusion, LIF is a simplified mathematical model based on the kinetic properties of biological neurons that is simple and good to use. The spiking neuron is the basic unit of calculation for SNNs. We use a unified spiking model to describe the actions of various spiking neurons as shown in Fig. 3, which includes the following discrete equations:

$$H[t] = f(V[t-1], X[t]) \qquad (1)$$

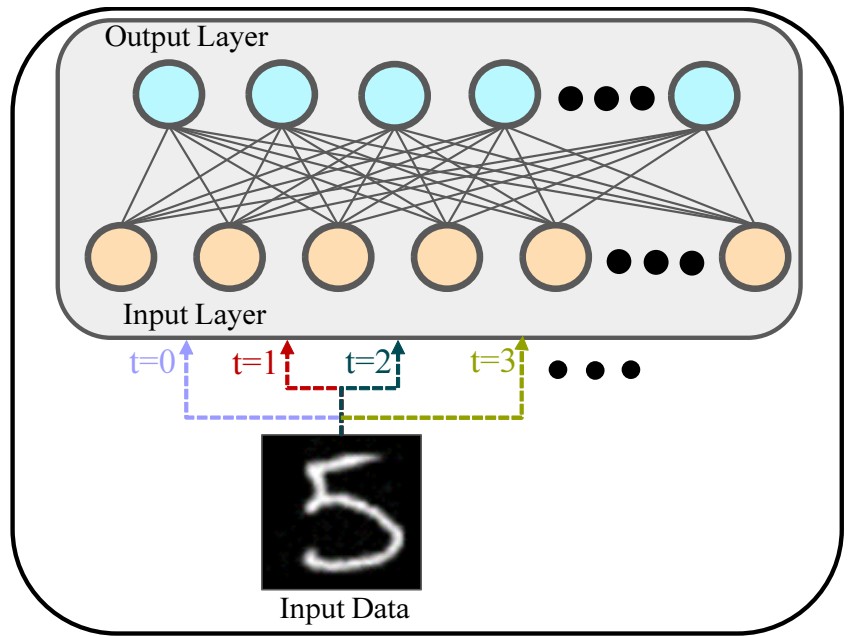

**Figure 2** **Frequency coding schematic.** An image of a handwritten digit ("5"), is passed through the input layer (bottom) to the output layer (top). The same image is processed at multiple time steps ($t = 0$, $t = 1$, $t = 2$, $t = 3$...), demonstrating the concept of frequency coding. At each time step, the input image is presented to the network, and its features are mapped to the output layer through weighted connections.

$$S[t] = \Theta(H[t] - V_{th}) \tag{2}$$

$$V[t] = H[t](1 - S[t]) + V_{reset}S[t]. \tag{3}$$

In the provided equation, $H[t]$ represents the input current at the specific time step $t$. Meanwhile, $H[t]$ and $V[t]$ respectively stand for the membrane potential following neuronal action and the membrane potential subsequent to a trigger pulse at time step $t$. The parameter $V$ denotes the triggering threshold, and $\Theta(x)$ corresponds to the Heaviside step function. This function is defined such that $\Theta(x) = 1$ when $x$ is greater than or equal to 0, and $\Theta(x) = 0$ when $x$ is less than 0. This formulation captures the essence of how the membrane potential and triggering mechanism interplay within the context of spiking neural networks.

$$\Theta(x) = 1 \text{ for } x \geq 0 \text{ and } \Theta(x) = 0 \text{ for } x < 0. \tag{4}$$

The $S[t]$ signifies the peak output at the specific time step $t$, adopting a value of 1 when a peak is present and 0 when it is not, thereby indicating the reset potential. This binary value captures whether a neuron has fired or not during that time step.

The function denoted as $f$ in Eq. (1) characterizes the neuronal dynamics and manifests distinct forms contingent on the specific spiking neuron models employed. For instance, in

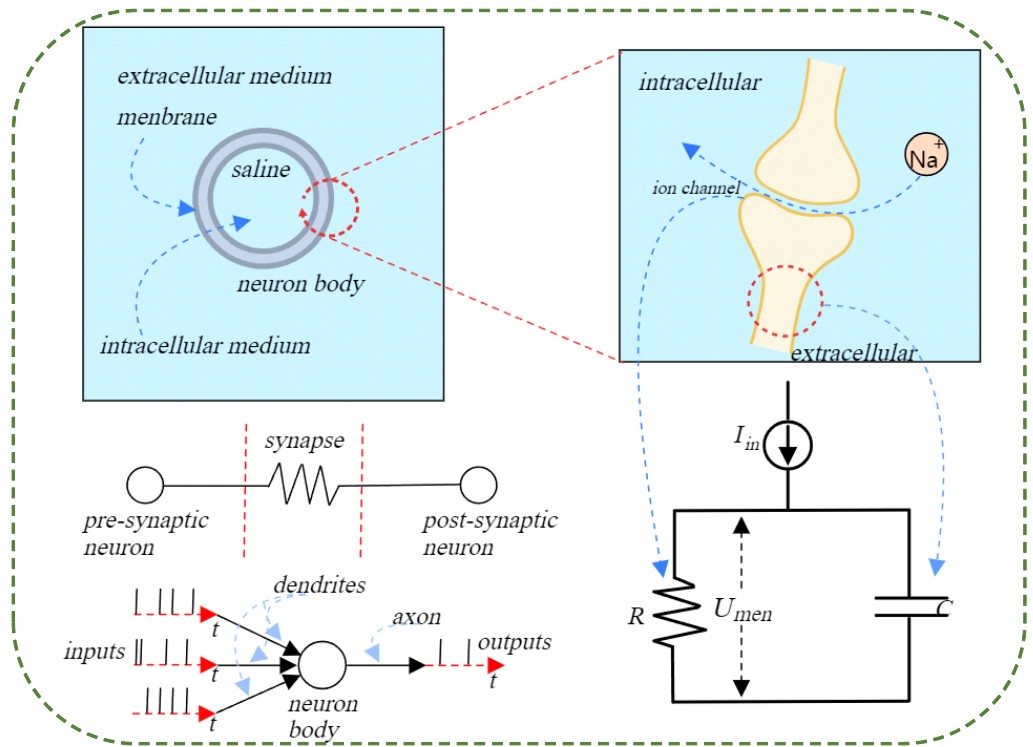

**Figure 3  Illustration of the action potential process in a spiking neuron.** When a neuron reaches a certain excitation threshold, an action potential is generated, propagates along the axon, and transmits the signal to downstream neurons. The left side shows the transmission process from presynaptic neuron to postsynaptic neuron, the center shows the dynamic balance of ions on both sides of the neuron membrane, and the right side is the equivalent circuit diagram, describing the change process of membrane potential with current input.

the Integrate and Fire (IF) model, the function $f$ can be mathematically represented using Eq. (5), while in the LIF model, it is expressed through Eq. (6). These distinct expressions for $f$ encapsulate the behavioral nuances of the respective spiking neuron models, shaping their responses to input stimuli and determining their spiking behavior characteristics.

$$H[t] = V[t-1] + X[t] \tag{5}$$

$$H[t] = V[t-1] + \frac{1}{\tau}X[t] - (V[t-1] - V_{\text{rest}}). \tag{6}$$

In the provided context, $\tau$ symbolizes the membrane time constant, which is a fundamental parameter in the equations being discussed. Equations (2) and (3) elucidate the mechanisms of pulse generation and the subsequent reset process, which are consistent across various types of pulse neuron models.

## Spiking residual neural networks

The residual block serves as a fundamental component within the architecture of ResNet, depicted in Fig. 4A. This figure showcases the elemental building block present in ResNet,

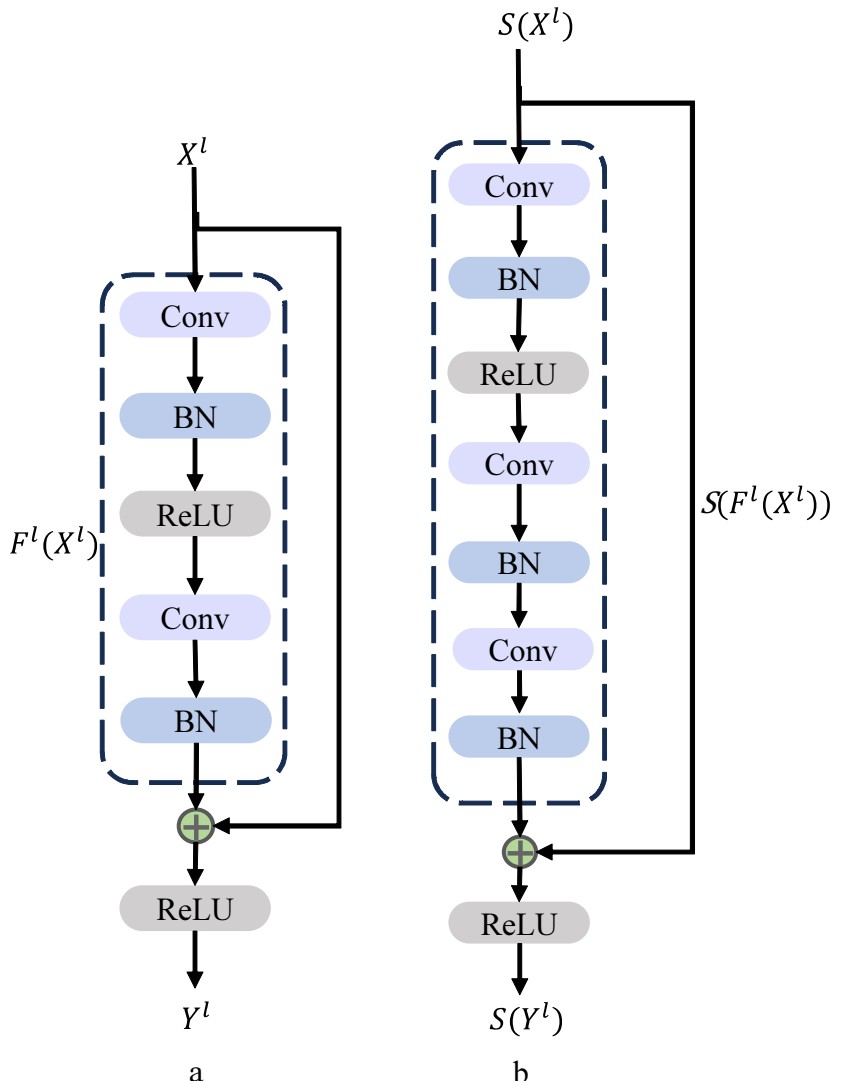

**Figure 4** **Residual block architecture. (A) Basic residual block architecture. (B) Improved residual block architecture.**

offering a visual representation of its structure and characteristics (*Han, Srinivasan & Roy, 2020*).

In the context described, $X^l$ and $Y^l$ respectively stand for the inputs and outputs of the initial block within the ResNet structure. The term "Conv" signifies the convolutional layer, while "BN" represents the batch normalization layer, and "ReLU" refers to the rectified linear unit activation layer. These components collectively constitute the elements that contribute to the operations and transformations occurring within the depicted ResNet block. The *b* in Fig. 4B is our improved spiking ResNet residual block, where $X^l$, $Y^l$ are the inputs and outputs of the first block in ResNet, Fig. 4B is our improved spiking ResNet residual block, which is based on the ResNet basic block with an additional layer of Conv and a layer of BN. The enhanced spiking ResNet demonstrates improved capabilities in

data feature extraction, effectively addressing challenges associated with gradient vanishing and gradient exploding due to heightened network depth. This augmentation facilitates more effective data feature extraction and successfully mitigates the concerns related to gradient vanishing and gradient exploding that tend to arise as network depth increases. We use residual learning for several stacked layers, and the constructed residual blocks are shown in Fig. 4B. In the formulation, we define the constructed block as:

$$\mathbf{y} = \mathcal{F}(\mathbf{x}, \{W_j\}) + \mathbf{x}. \tag{7}$$

The $x$ and $y$ respectively refer to the inputs and outputs of the specific layer under consideration. The function $\mathcal{F}(\mathbf{x}, W_i)$ signifies the residual mapping that the network aims to learn. In the two-layer example depicted in Fig. 4, the formulation $\mathcal{F} = W_2\sigma(W_1\mathbf{x})$ is employed, where $\sigma$ represents a simplified version of the rectified linear unit (ReLU) function along with biases. The operation $\mathcal{F} + \mathbf{x}$ is executed through a combination of shortcut concatenation and elementwise addition. Following the addition, a nonlinear operation (namely, $\sigma(y)$) is applied to the result, contributing to the network's overall transformation process.

The shortcut connection defined in Eq. (7) exhibits the advantageous qualities of not introducing an increase in parameters or additional computational complexity. This characteristic holds substantial practical appeal and proves especially significant when comparing ordinary networks to their residual counterparts.

This property becomes notably significant when comparing networks with equal parameters, depth, width, and computational costs (with minimal computational impact due to elementwise addition) between ordinary and residual networks. It's essential to note that for the dimensions of $x$ and $\mathcal{F}$ to align precisely, enabling effective shortcut concatenation, in cases where this alignment is not naturally achieved, such as when there are alterations in input and output channels, a linear mapping can be employed through the shortcut connections to ensure the appropriate dimension matching. This process guarantees seamless integration and effective utilization of the residual connections without introducing additional parameters or computational overhead.

The structure of the residual function is adaptable and offers flexibility. The experiments conducted in this paper encompass a function comprising three layers, as depicted in Fig. 4. While the notations outlined earlier pertain to fully connected layers for the sake of simplicity, they are equally applicable to convolutional layers. The functional representation can indeed encompass multiple convolutional layers.

In this context, when dealing with convolutional layers, the process is performed channel by channel. Specifically, the elements are summed across the two feature maps, contributing to the generation of the final outcome. This mechanism allows for the integration of information and the preservation of relevant features throughout the network's architecture, thus contributing to its enhanced performance.

### SECA module for channel attention

Traditional attention mechanisms, such as SENet, allocate weights between channels through fully connected layers. This process introduces a large number of parameters,

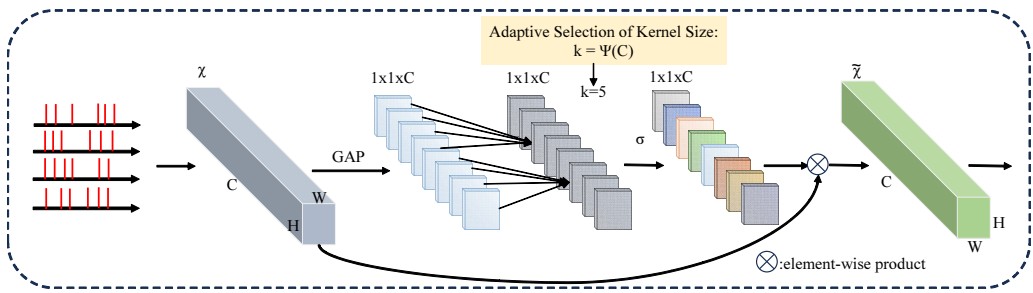

**Figure 5** **Spiking-Efficient Channel Attention (SECA) module architecture.** GAP is used for global spatial information aggregation, SECA generates channel weights by performing fast 1D convolutions of size k, where k is adaptively determined by mapping the channel dimension C.

especially when processing high-dimensional feature maps, which increases computational complexity. In order to solve this problem, we propose a method based on local cross-channel interaction, it refers to local interactions between different feature channels rather than capturing attention weights through global channel relationships. Typically in a convolutional neural network, each feature map (channel) processes the input independently. Cross-channel interaction means that channels are no longer completely independent, but influence each other. The Spiking-Efficient Channel Attention (SECA) mechanism calculates the relationship between channels through local interactions without introducing additional parameters or complex global pooling. The core innovation of SECA is that it introduces a parameter-free local one-dimensional convolution to realize the interaction between channels, simplifying the calculation of the attention mechanism while maintaining the ability to identify important features.

Following a thorough examination of the impacts stemming from channel dimension reduction and cross-channel interaction within the realm of channel attention learning, the conclusions drawn from these analyses lay the foundation for the introduction of the SECA module (*Wang et al., 2020*). The architecture of this module is visually depicted in Fig. 5, illustrating how the insights garnered from the earlier analyses are translated into a concrete design aimed at optimizing channel attention mechanisms.

When presented with an aggregated feature denoted as $\mathbf{y} \in \mathbb{R}^C$, where $C$ represents the channel dimension, the channel information can be propagated or carried forward through the network architecture.

$$\omega = \sigma(\mathbf{Wy}). \tag{8}$$

In this context, $\mathbf{W}$ represents a parameter matrix with dimensions $C \times C$. Within this paper, an approach is employed to capture local cross-channel interactions that emphasize efficiency and efficacy. Specifically, a band matrix denoted as $\mathbf{W}_k$ is utilized to learn channel

attention. This band matrix $\mathbf{W}_k$

$$
\begin{bmatrix}
w^{1,1} & \cdots & w^{1,k} & 0 & 0 & \cdots & \cdots & 0 \\
0 & w^{2,2} & \cdots & w^{2,k+1} & 0 & \cdots & \cdots & 0 \\
\vdots & \vdots & \vdots & \vdots & \ddots & \vdots & \vdots & \vdots \\
0 & \cdots & 0 & 0 & \cdots & w^{C,C-k+1} & \cdots & w^{C,C}
\end{bmatrix}
\tag{9}
$$

encompasses $k \times C$ parameters, as evident in Eq. (9). This matrix is typically relatively small in size. Moreover, Eq. (8) avoids complete isolation between distinct groups of interactions. Equation (9) considers only the interactions between $y_i$ and its " $k$" neighbors, effectively capturing local dependencies. This approach fosters a balance between preserving relevant information within the local context and maintaining computational efficiency.

$$
\omega_i = \sigma\left(\sum_{j=1}^{k} w_i^j y_i^j\right), y_i^j \in \Omega_i^k.
\tag{10}
$$

Here, $\Omega_i^k$ signifies the collection of " $k$" neighboring channels associated with the channel $y_i$. This set encapsulates the specific subset of nearby channels that contribute to the local cross-channel interactions involving the particular channel $y_i$.

A more efficient approach would be to have all channels share the same learning parameters, *i.e.*:

$$
\omega_i = \sigma\left(\sum_{j=1}^{k} w^j y_i^j\right), y_i^j \in \Omega_i^k.
\tag{11}
$$

Note that this strategy can be easily realized by fast 1D convolution with kernel size $k$, *i.e.*,

$$
\omega = \sigma(\mathrm{C1D}_k(\mathbf{y})).
\tag{12}
$$

In this context, C1D refers to a 1D convolution operation. The approach outlined in Eq. (12) is implemented through the utilization of the SECA (spiking-efficient channel attention) module, characterized by only $k$ parameters. Within this module, the chosen value is $k = 3$, a decision made to ensure the optimal balance between model complexity and the capability to effectively capture local cross-channel interactions (*Wang et al., 2020*). The choice of $k = 3$ enhances efficiency while maintaining the effectiveness of the module. The PyTorch implementation is shown below:

Listing 1: SECA Attention

```
avg_pool = AdaptiveAvgPool2d(1)
conv = Conv1d(1, 1, kernel_size=kernel_size,
              padding=(kernel_size - 1) // 2, bias=False)
sigmoid = Sigmoid()

def forward(x):
    y = avg_pool(x)
```

```
    y = conv(y.squeeze(-1).transpose(-1, -2))
    y = y.transpose(-1, -2).unsqueeze(-1)
    y = sigmoid(y)
    return x * y.expand_as(x)
```

We adopt a local cross-channel interaction strategy, but only one channel is used in the experiments. The reason for choosing this approach is to simplify the analysis of cross-channel interactions, allowing us to focus on fundamental aspects of the interaction without being distracted by the complexity of multiple channels. The image classification task targeted in this article has been proven to achieve remarkable results under single-channel input.

Given our incorporation of the efficient channel attention (ECA) module (*Wang et al., 2020*), a meticulously designed construct adept at effectively capturing localized cross-channel interactions, the necessity to assess the scope of these interactions (indicated by the kernel size $k$ in the context of 1D convolution) becomes imperative. Across diverse CNN architectures (*O'Shea & Nash, 2015*), the optimum span of interaction for cross-validation purposes can be subject to manual adjustments, aimed at accommodating convolutional blocks characterized by varying channel quantities. Nonetheless, this manual fine-tuning procedure, executed *via* cross-validation, has the potential to impose substantial computational demands.

It is noteworthy that the application of group convolutions has yielded significant enhancements to CNN architectures. In these configurations, channels of high dimensionality (or low dimensionality) are subjected to convolutional operations spanning expansive (or restricted) domains, contingent on a predetermined quantity of groupings. Drawing upon analogous concepts, a plausible supposition emerges: the extent of interaction, denoted by the kernel size $k$ in 1D convolution, might demonstrate a correlation with the channel dimension $C$. This correlation is conceptualized through a functional relationship denoted as $\varphi$, which potentially delineates the connection between $k$ and $C$:

$$C = \phi(k). \tag{13}$$

This functional expression $\varphi$ encapsulates the dynamics by which the optimal kernel size $k$ evolves in congruence with fluctuating channel dimensions. This construct furnishes a framework for adapting interaction ranges to distinct architectural contexts, obviating the necessity for extensive manual adaptations and resource-intensive cross-validation procedures.

The most basic form of mapping involves a linear function, expressed as $\varphi(k) = \gamma k - b$. Nevertheless, the scope of this relationship is notably constrained within the confines of a linear function. Conversely, it is widely acknowledged that the channel dimension $C$—also referred to as the number of filters—is often configured as a power of two. Consequently, an alternative avenue is introduced by imbuing the linear function $\varphi(k) = \gamma k - b$ with nonlinearity. This step aims to leverage the inherent characteristics of power-of-two

channel dimensions to enhance the expressiveness and adaptability of the relationship.

$$C = \phi(k) = 2^{(\gamma * k - b)}. \tag{14}$$

Subsequently, when presented with the channel dimension $C$, the appropriate kernel size $k$ can be dynamically ascertained through an adaptive process by:

$$k = \psi(C) = \left| \frac{log_2(C)}{\gamma} + \frac{b}{\gamma} \right|_{odd}. \tag{15}$$

Here, $|t|_{\text{odd}}$ signifies the nearest odd number to the value of $t$. In the context of this research, we uniformly assign the values of $\gamma$ and $b$ as 2 and 1, respectively, across all experimental scenarios. Evidently, employing this mapping strategy results in a situation where high-dimensional channels experience interactions spanning greater distances, whereas low-dimensional channels engage in interactions spanning shorter distances. This outcome is achieved by means of the nonlinear mapping, which optimally adapts the kernel size according to the channel dimension, thereby enhancing the efficacy of cross-channel interactions.

Although ResNet's residual connection effectively compensates for the loss function problem in deep networks, the SECA attention mechanism further improves feature extraction capabilities. By adaptively adjusting the weight of the channel, SECA emphasizes the important information in the input features, allowing the model to have better expression ability and generalization performance when facing complex data. In our series of experiments, the loss function of the model was significantly reduced after adding SECA, verifying the necessity of SECA in optimizing network performance and improving feature selection.

### Spiking residual attention neural networks

The basic architecture of the SNNs model consists of a data input layer, residual blocks, LIF neurons, an attention mechanism (SECA), and a fully connected layer. The inputs to the model are the dataset of image classes and the DVS dataset, and the outputs are the classes of the target. The input layer receives the image to be categorized, and after passing through the spiking neurons, the image is encoded into a sequence of spikings with a certain time step. The pulse sequence is propagated forward between the layers, and the pulse sequence generated in the pulse output layer is accumulated over the time step and used to achieve the final classification decision. The spiking residual attention neural network designed in this work is shown in Fig. 6. The final structure of our model is composed of five residual blocks, three LIF neurons, a SECA module, a dropout layer, and a fully connected layer. The structure of the model is the best position that we have obtained through our experiments.

In this paper, we introduce LIF neurons between different ResNet blocks to enhance the transfer and integration of information. LIF neurons effectively maintain the flow of information on important features through the spiking mechanism, thereby alleviating the vanishing gradient problem in deep networks. In addition, LIF neurons are able to capture temporal dependencies through dynamic spike sequences when processing time series data,

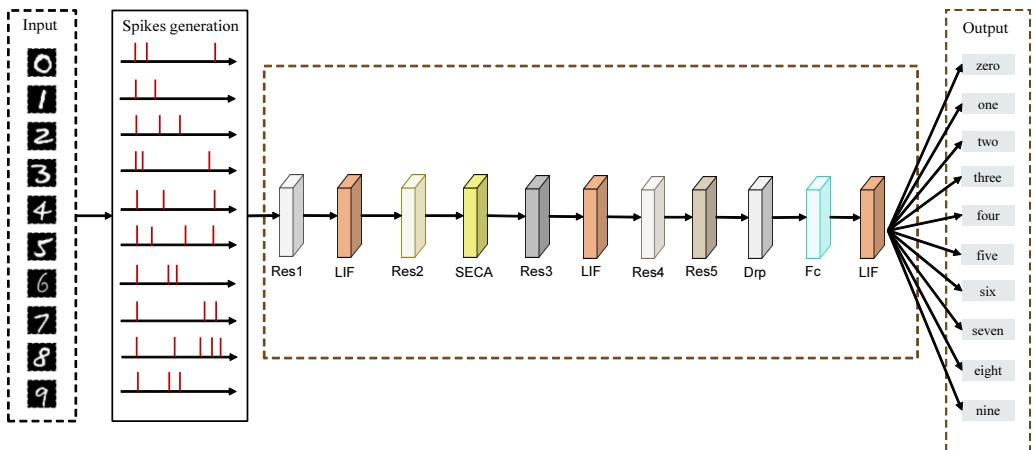

**Figure 6 Spiking Residual Attention Neural Networks architecture.** The input is an image of handwritten digits, and the image signal is converted into a pulse sequence through the pulse generation module, which passes through the residual block, LIF, SECA, and Fc in sequence. The final output layer generates corresponding classification results for identifying handwritten digits 0–9.

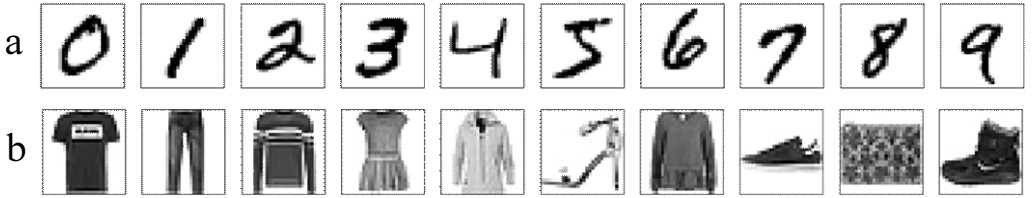

**Figure 7 Illustrative images of the dataset.** (A) A example from the MNIST dataset; (B) an example from the Fashion-MNIST dataset.

further improving the model's performance in complex tasks. Experimental results show that the network using LIF neurons has achieved significant performance improvements in image classification, verifying its potential in deep learning.

# EXPERIMENTAL EVALUATION

In this section, we evaluate the proposed Sa-SNN model on three benchmark datasets: MNIST, Fashion-MNIST, and N-MNIST. To further demonstrate the effectiveness of our proposed model, we add the SECA model to different locations within the network for a series of comparisons. All experiments are conducted on a computer with a 3.6 GHz Intel (R) CoreTM i7-12700KF processor and an NVIDIA GeForce GTX 3090 graphics card with 24G of memory. The deep learning framework used is PyTorch.

## Experimental datasets

We chose three datasets with publicly available data to evaluate our proposed model and compare it with other models. Figure 7 shows some images of each dataset, and Table 1 shows an overview of their input, training, and test set sizes.

**Table 1  Overview of datasets.** The input dimension, number of training set, and number of testing set for three publicly available datasets.

| Dataset | Input size | Training | Testing |
|---------|-----------|----------|---------|
| MNIST | $28 \times 28 \times 1$ | 60,000 | 10,000 |
| Fashion-MNIST | $28 \times 28 \times 1$ | 60,000 | 10,000 |
| N-MNIST | $34 \times 34 \times 2$ | 60,000 | 10,000 |

The initial dataset under consideration is the MNIST dataset, a widely adopted image classification dataset. The dataset comprises grayscale images depicting handwritten numerical digits ranging from "0" to "9". The dataset is divided into two subsets: a training set consisting of 60,000 samples and a test set containing 10,000 samples. Because MNIST datasets are usually pre-partitioned, this division ensures the standardization of evaluation models, so that different researchers can make fair comparisons in different experiments.

Comprising a total of 60,000 training samples and 10,000 test samples, the Fashion-MNIST dataset serves as another subject of investigation. This dataset involves grayscale fashion clothing images, each sized at $28 \times 28$ pixels, which is comparable to the original MNIST input dimensions. The dataset encompasses ten distinct categories, including t-shirts/tops, pants, pullovers, dresses, jackets, sandals, shirts, sneakers, bags, and boots.

The Neuromorphic-MNIST (N-MNIST) dataset is a spiking variant derived from the conventional frame-based MNIST dataset. It encompasses an identical composition of 60,000 training samples and 10,000 test samples, aligning with the scale of the original MNIST dataset at $34 \times 34$ pixels. The N-MNIST dataset is generated by employing the ATIS sensor affixed to a motorized gimbal, which undergoes movement while observing MNIST examples displayed on an LCD monitor. This approach introduces spiking neural network-relevant dynamics to the dataset, rendering it conducive to investigating the performance of spiking neural networks.

## Ablation experiments

In this section, we will present the effects brought by the SECA module at different locations in our network model and whether there are any effects on the network. The results are shown in Fig. 8.

First of all, we conducted numerous experiments on the experimental platform to verify the effectiveness of our proposed SECA module. These results demonstrate that our proposed SECA module has a significant impact on accuracy. For instance, the recognition accuracy of Fashion-MNIST before adding the SECA module is 93.21%, and after the addition of the SECA module, the accuracy improves to 94.13%, indicating a 0.92% enhancement in accuracy.

Secondly, we carried out a series of experiments to investigate the effects of placing the SECA module at different positions within our network model. We initiated experiments by positioning the SECA module at the network's forefront and subsequently altering the SECA module's location incrementally until the highest accuracy was achieved, while

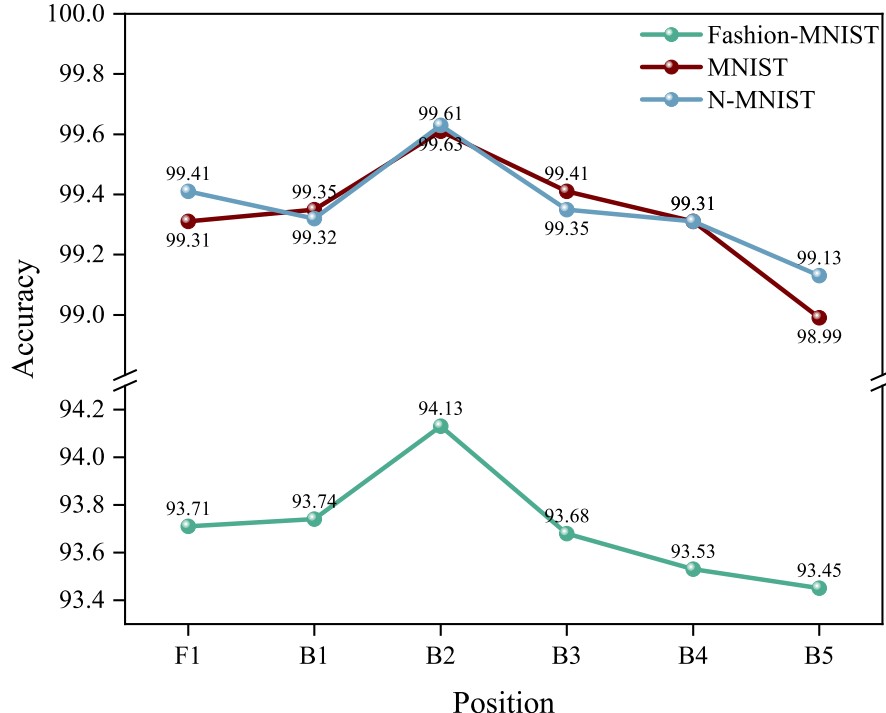

**Figure 8 Accuracy of SECA modules at different positions.** "F1" represents the SECA module located before the first layer, and "B1" represents the SECA module located after the second layer.

keeping the other network structures constant. The outcomes of these experiments are illustrated in Fig. 8.

From the experimental results, it can be seen that the best recognition accuracy of 94.13% is achieved when the SECA module is located after the second layer of residual blocks.

## Comparison tests

With increasing research, many spiking neural network structures for image classification tasks have emerged in recent years. Table 2 compares the structure proposed in this paper with other studies based on the MNIST dataset.

As shown in Table 2, the spiking RBM network architecture is not very effective in learning features, while the recognition accuracy of Sa-SNN is 99.61%. By incorporating the SECA attention mechanism in the network architecture, the problem of information loss is effectively compensated. Therefore, the comparison with other network architectures indicates that the method proposed in this paper achieves the best performance without the need for data preprocessing.

In addition, the effectiveness of the structure is verified on the Fashion-MNIST dataset, and the experimental results are compared with the results of related studies. The results are shown in Table 3. The deep SNN of the TSSL-BP algorithm is proposed, which contains a total of two convolutional layers, two pooling layers, and one fully connected

**Table 2  Classification accuracy comparison with different SNNs on MNIST.**

| Network-type | Preprocessing | Accuracy |
|---|---|---|
| STDP-trained network (*Thiele, Bichler & Dupret, 2018*) | None | 93.50% |
| Feed forward network | Edge-detection | 96.50% |
| Spiking ConvNet | Thresholding | 91.29% |
| Dendritic neurons | Thresholding | 90.26% |
| Spiking RBM | Thresholding | 91.90% |
| Spiking RBM | Enhanced trained set | 94.09% |
| AutoML-SNN (*Roy et al., 2024*) | None | 99.43% |
| STB-STDP (*Dong et al., 2023*) | None | 97.90% |
| ANN-SNN (*Han et al., 2024*) | None | 92.33% |
| STDBP | Temporal coding | 98.10% |
| D-SNN | Thresholding | 98.47% |
| Sa-SNN (ours) | None | 99.61% |

**Table 3  Classification accuracy comparison with different SNNs on Fashion-MNIST.**

| Dataset | Model | Accuracy |
|---|---|---|
| Fashion-MNIST | S4NN (*Kheradpisheh & Masquelier, 2020*) | 88.00% |
| | TSSL-BP (*Zhang & Li, 2020*) | 92.83% |
| | Phase-SNN (*Cai et al., 2024*) | 89.60% |
| | CSNN (*Viet Ngu, 2024*) | 84.23% |
| | BS4NN (*Kheradpisheh & Masquelier, 2020*) | 87.30% |
| | STDBP (*Zhang et al., 2021*) | 90.10% |
| | Sa-SNN (ours) | 94.13% |

layer. However, the network structure in this paper has five residual blocks and one fully connected layer. The recognition accuracy increases from 92.83% to 94.13% while increasing the network depth. Compared with deep neural networks, the network structure proposed in this paper also shows significant improvement in recognition accuracy.

## Experimental results

First, we conducted experiments on different datasets. For the MNIST, Fashion-MNIST, and N-MNIST datasets, the experimental results are shown in Table 4. The recognition rate reaches 99.61%, 94.13%, and 99.53%, respectively.

For different datasets, Sa-SNN always achieves excellent performance. the advantages of Sa-SNN in image classification and information acquisition are obvious. It is more suitable for deployment and application on devices with limited resources. It also demonstrates that the method proposed in this paper has good generalization ability.

## CONCLUSIONS

In this paper, an efficient deep SNN structure, Sa-SNN, is designed for image classification; an improved residual structure is proposed for spiking neural networks; and the SECA channel attention mechanism is incorporated as a compensation mechanism to make up

**Table 4  Performance comparison of different datasets on different models.**

| Dataset | Model | Accuracy |
|---------|-------|----------|
| MNIST | Mostafa (*Mostafa, 2017*) | 97.50% |
| | Cosma (*Comsa et al., 2020*) | 97.90% |
| | S4NN (*Kheradpisheh & Masquelier, 2020*) | 97.40% |
| | BS4NN (*Kheradpisheh & Masquelier, 2020*) | 97.00% |
| | STiDi-BP (*Mirsadeghi et al., 2021*) | 97.40% |
| | TSSL-BP (*Zhang & Li, 2020*) | 99.53% |
| | STDBP (*Zhang et al., 2021*) | 99.40% |
| | **Sa-SNN (ours)** | **99.61%** |
| N-MNIST | TSSL-BP (*Zhang & Li, 2020*) | 99.40% |
| | **Sa-SNN (ours)** | **99.53%** |

**Notes.**

The proposed model is shown in bold.

for the loss of information. By comparing it with a variety of related methods, Sa-SNN shows a significant improvement in recognition accuracy. In addition, Sa-SNN has a relatively simple network structure, containing only convolutional residual blocks and spiking neurons.

The SECA model proposed in this paper compensates for the shortcomings of the traditional loss function through its specific design. By adjusting the model parameters, the SECA model effectively reduces the error rate while maintaining the convergence of the loss function. Experimental results show that the SECA model performs significantly better than existing models on different tasks, especially when processing high-dimensional data, showing better robustness and generalization capabilities. The introduction of this model significantly improves the performance of the overall system and verifies its potential in practical applications.

In the future, we will apply our network structure to more spiking neural networks and further investigate the combination of SECA and spatial attention modules.

## ACKNOWLEDGEMENTS

We express our gratitude to the authors of the MNIST dataset, the Fashion-MNIST dataset, and the N-MNIST dataset for generously providing open data sets to support our research endeavors.

### Funding

This research is supported by the Henan Provincial Science and Technology Research Project (242102210006). The funders had no role in study design, data collection and analysis, decision to publish, or preparation of the manuscript.

### Grant Disclosures

The following grant information was disclosed by the authors:

The Henan Provincial Science and Technology Research Project: 242102210006.

## Competing Interests

The authors declare there are no competing interests.

## Author Contributions

- Yongping Dan conceived and designed the experiments, performed the experiments, performed the computation work, prepared figures and/or tables, authored or reviewed drafts of the article, and approved the final draft.
- Zhida Wang conceived and designed the experiments, performed the experiments, performed the computation work, prepared figures and/or tables, and approved the final draft.
- Hengyi Li analyzed the data, authored or reviewed drafts of the article, and approved the final draft.
- Jintong Wei analyzed the data, prepared figures and/or tables, authored or reviewed drafts of the article, and approved the final draft.

## Data Availability

The code and raw data are available in the Supplemental Files.

The third party data is available at:

- MNIST Dataset: https://www.kaggle.com/datasets/hojjatk/mnist-dataset.

- Fashion-MNIST dataset: https://www.kaggle.com/datasets/zalando-research/fashionmnist.

- N-MNIST: https://www.garrickorchard.com/datasets/n-mnist.

## Supplemental Information

Supplemental information for this article can be found online at http://dx.doi.org/10.7717/peerj-cs.2549#supplemental-information.

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
