# Peer review of "Sa-SNN: spiking attention neural network for image classification"

_PeerJ Computer Science, doi:10.7717/peerj-cs.2549_

## Round 0.1 · original submission · Major Revisions

After reviewing the opinions, the decision to request substantial revisions is justified by the fact that the manuscript, in its current form, presents points of confusion in its conceptual and methodological explanation, as well as essential omissions in the comparison with the state of the art. Although the work has merit and makes an exciting contribution, it needs significant improvements to achieve the level of clarity and depth expected.

The reviewers have raised pertinent points that, once addressed, will considerably strengthen the article. I therefore encourage the authors to make the suggested revisions and submit a new version for re-evaluation.

The most important points are:

Conceptual clarity and detailed explanations: The concept of 'Sa-SNN' and the functioning of the SECA module are central to the article's proposal but are insufficiently described. The explanation of these terms, particularly in the summary and introduction, must be more direct. The lack of clarity on these points weakens the overall understanding and impact of the proposal.

Comparison with state-of-the-art: The reviewers emphasized updating the comparison with more recent methods. The article makes comparisons with older work, but it is crucial to include recent research (such as Han et al., 2024 and Roy et al., 2024) to correctly position the article's contribution in the current context of the field.

Diagramming and visualization: The absence of diagrams illustrating the architecture of the proposed model makes it difficult for readers to understand the innovation of the work. Diagrams explaining the workflow, the SECA architecture, and its integration with the model are essential.

Documentation and code: The code provided is disorganized and lacks adequate documentation. Reviewers and future readers will find it difficult to replicate or analyze the results without a detailed explanation of the code files and their functions. Clear documentation is imperative to increase the transparency and replicability of the research.

In addition, since revisions will be made, I request:

There is a lack of justification for methodological choices: The article lacks a more in-depth discussion of the methodological choices made in developing SECA and its integration with ResNet. The reasoning behind these choices is unclear and should be explained to strengthen the proposal.

It is crucial to demonstrate performance compared to recent methods: Apart from including recent work in the comparison, the article does not sufficiently address the impact of its model on state-of-the-art SNN networks. A more detailed explanation of why SECA stands out compared to other attention approaches would be beneficial.

Reviewer 1 ·

Basic reporting

Strengths:
The article presents a clear motivation for the research, i.e., the biological plausibility of spiking neural networks (SNNs) and the energy efficiency advantage over conventional neural networks. It also introduces the Spiking-efficient channel attention (SECA) module and briefly describes the experimental results. Also, the experimental design and follows standard benchmarks in the field (MNIST, Fashion-MNIST, and N-MNIST datasets). The results reported are particularly with the consistent improvement over baselines.

Weaknesses
● Some parts of the text lack clarity, especially around key contributions. For instance, the sentence "Sa-SNN can be more easily implemented for spiking neural network training" is vague. What exactly makes it easier to implement, and how is it measured?
● The use of terms like "Sa-SNN" is not properly explained in the abstract. It’s unclear what this specific term means until the reader reaches the end?
● The description of the SECA module lacks depth. While it's clear that it’s inspired by the attention mechanism, it doesn’t explain how it improves spiking networks or what its main advantages are over existing methods in detail.
● Also, there is no mention of how the SECA module compares to other attention mechanisms or why dimensionality reduction is avoided, which could be a key differentiating factor but is not elaborated on. “Clarify what "Sa-SNN" stands for earlier in the abstract.”
● The energy efficiency of the proposed model is not quantified. Given that one of the key advantages of SNNs is low power consumption, the paper could strengthen its claims by providing a comparison of energy efficiency between Sa-SNN and other models.
● The manuscript needs more polishing in terms of language and formatting. A more thorough revision of the writing would improve clarity and presentation.
● The authors mention, 'In this study, we will introduce related work on Spiking Neural Networks (SNNs) in Section 2. In Section 3, we will provide a detailed overview of the implementation methods of SNNs. Specific implementation details will be emphasized in Section 4. Finally, Section 5 will summarize the entire paper.' However, the paper does not contain numbered sections. The authors need to either include section numbers or remove these references throughout the paper.
● Also, the authors frequently use different abbreviations repeatedly for example, 'SNN' for Spiking Neural Networks.
● Line 140. Spike encoding? What does the reader consider this, a typo mistake or a sub section? Similar in 168, 206……..
● In the first 10 pages of the paper, the authors provide a thorough explanation of general concepts related to Spiking Neural Networks, including the Spiking Residual Attention Neural Networks, SECA Module for Channel Attention, Spiking Residual Neural Networks, and Spiking neuron models, supported by general conceptual figures. However, the paper lacks specific diagrams or flowcharts depicting the authors' own workflow or architectural design. Given that these details are very important for understanding the novel aspects of the proposed model, why have the authors not included such diagrams or detailed architectural representations of their own work? Including these would enhance clarity and provide readers with a clearer understanding of the proposed methods and their implementation.
● The authors compare their results with methods from several years ago and mention that their approach is state-of-the-art. However, to substantiate their claim of advancing beyond current state-of-the-art methods, they should also compare their results with the latest state-of-the-art methods in the field. For example, comparison with these recent research studies,
1-Han, Yanan, et al. "Conversion of a single-layer ANN to photonic SNN for pattern recognition." Science China Information Sciences 67.1 (2024): 112403.
2-Roy, Kaushik, Ulrich Ruckert, and Thorsten Jungeblut. "Poster: Selection of Optimal Neural Model using Spiking Neural Network for Edge Computing." 2024 IEEE 44th International Conference on Distributed Computing Systems (ICDCS). IEEE, 2024.
3-Mukhoty, Bhaskar, et al. "Direct training of snn using local zeroth order method." Advances in Neural Information Processing Systems 36 (2023): 18994-19014.
4-Dong, Yiting, et al. "An unsupervised STDP-based spiking neural network inspired by biologically plausible learning rules and connections." Neural Networks 165 (2023): 799-808.
This comparison would provide a clearer context for evaluating the performance and novelty of their proposed method.

Minor Mistakes
● There are several typographical errors, e.g., “MNIST,FASHION-MNIST,N-MNIST, datasets” --> “MNIST, Fashion-MNIST, and M-MNIST datsets”. This gives the abstract an unpolished feel.
● It seems the authors missed the opportunity to highlight their specific contributions in the introduction. This is a common practice in scientific papers, as it helps readers understand what is new and unique about the work. it would strengthen the paper if the authors added a concise contribution summary at the end of the introduction to clarify what they bring to the field.
● While the references are generally well-chosen, there could be additional discussion on recent advancements in SNNs and attention mechanisms to better position this work within the state of the art.
● Some technical terms could benefit from more context. For example, when introducing key concepts like "local cross-channel interaction" and "frequency coding," it would help to explain these terms more clearly and how they differ from other coding schemes.
● Authors need to cite recent state-of-the-art research studies.

My key observations are as follows:

● Lack of Documentation: The code provided is divided into multiple files without clear labeling or documentation. Reviewers do not have the time to understand the flow and methodology on their own. It is essential that detailed documentation be provided, including the purpose of each file, code flow, and usage instructions.
● Comparison with State-of-the-Art Methods: The manuscript does not adequately compare the proposed work with state-of-the-art methods. The comparison provided is only with methods from several years ago. A comprehensive comparison with current, state-of-the-art methods is crucial for evaluating the significance and impact of the proposed approach.

Experimental design

Comments on Provided Code:
• File- compare,2,3,4.py files:
o Why are the transforms.Resize and transforms.Grayscale commented out in your transforms? Are they not needed for your specific task?
o The ECAAttention module is defined but not used in the Net class. Was it intended to be part of the network architecture? If so, where should it be integrated?
o Why is there a commented-out conv3 and bn3 in ResidualBlock? Are they experiments or leftovers from previous versions?
o The forward_pass function is commented out. Are there specific reasons for this, or is it planned to be used later?
o The learning rate is set to 0.0001. Is this value chosen based on experimentation, or is it a default?
o You have defined loss_fn = SF.ce_rate_loss() but also commented out nn.CrossEntropyLoss()
o Dropout is applied only after the spk2 in the forward pass. Is this placement optimal for your network’s training and performance?
o The training loop is implemented with a fixed learning rate of 0.0001 and updates every 50 iterations. Is this learning rate optimal, and is 50 iterations a suitable interval?
• File- ECA_model.py:
o How is the kernel size for the convolution layer in the ECAAttention module determined? And Why is it important that the kernel size is an odd number?
o What is the purpose of using nn.AdaptiveAvgPool2d(1) before applying the convolution? And Why is self.conv applied to the pooled output (y) rather than directly to the input tensor (x)?
o Minor: How do the parameters gamma and b affect the attention mechanism in the ECAAttention module?
• File- eca-scnn.py:
o Why is transforms.Resize((28, 28)) used even though FashionMNIST images are already 28x28 pixels? And What is the effect of transforms.Grayscale() on the data, and why is it used here?
o Why might the normalization parameters differ from those in the comments (transforms.Normalize((0.1307,), (0.3081,)))?
o How does nn.Leaky with beta=0.5 function compared to other types of Leaky Integrate-and-Fire (LIF) neurons? And why author used Batch Normalization layers after the convolutional layers?
o How does the network architecture (with two convolutional layers and a fully connected layer) contribute to the performance on FashionMNIST? Even though only two pooling operations used in the network?
• The code appears to be divided into multiple files without clear labeling or documentation, making it challenging to understand the flow and purpose of each component. Given that reviewers do not have time to first understand the flow and then explore the working methodology, could you provide detailed documentation that outlines:
o Purpose of Each File: What each file is responsible for.
o Code Flow: How the files and functions interact.
o Usage Instructions: How to run and configure the code
• This will greatly assist in understanding and evaluating the code effectively.

Validity of the findings

no comment

Additional comments

In the current state, I suggest a weak reject.
AFTER the REVISION of the above mentioned issues I would suggest an accept.

Cite this review as

·

Basic reporting

The authors have written the article well in English with very few spelling and grammatical mistakes.
Figures and tables
1. Most of the figures are well structured but the legends are poorly written so it has to be rewritten to bring out the true meaning of the figure.
2. Table titles and legends are usually placed on top of the table
3. Fig 1 legends need to be fixed as it is not readable in the current form like ‘simple’ might be ‘sample
4. In Fig 4 the notations are not clear like GAP, K=5 etc which needs to be clearly mentioned in the legends
5. Why was N-MNIST not shown in Fig 6
Raw data and code
1. There is a missing readme in the set of codes which makes it difficult to which is for what
2. Some of the comments seem to be in Chinese language which makes it difficult for others to understand
Language check
1. Ln13,14,15- space must be provided at each statement's beginning.
2. Ln 16- ‘a’ instead of ‘an’
3. Ln 142- “ imspikes” I think is a spelling mistake
4. Ln 178- New statement should start with a capital letter
5. Ln185- extra space in between and
6. Ln201-205 is not at all in the readable form. It has to be either deleted or rephrased completely
The review of literature seemed a bit dull with missing papers on bioinspired spiking networks For e.g., In Ln 64 Bioinspired deep layered spiking neural network has already come up before DSNN. I recommend adding citations to works such as Vijayan & Diwakar (2022) and the papers from Egidio D'Angelo's group. This will add depth and credibility to their review of the field.

Experimental design

The authors have proposed a Spiking-efficient channel attention (SECA) module that employs a local cross-channel interaction strategy so that the need for dimensionality reduction can be avoided and also boost the performance of the network. The methods on spike encoding, neuron model, SNN, ResNet, SECA and the network model has been explained.

Validity of the findings

The novelty is the SECA model which the authors claim would compensate for the loss function. The conclusion does not capture that to the full extent.
1. There is no statistical evidence that it is not rote learning.
2. The Resnet itself would compensate for the loss function then what is the need of SECA.
3. The role of LIF in between different Resnet blocks is also not clear.
4. When it was suggested that a local cross-channel interaction strategy is used, it was only 1 channel that was used. There should be a statement that Justify this also in the paper
5. The time complexity required for the computations is not stated anywhere in the paper

Additional comments

1. Ln 40-51-The comparison between the DNN and SNN and the motivation for having SNN instead of DNN remain vague
2. Ln114-115 reference to this line is important and missing
3. Ln-146- The authors have talked about the 3 S but have showcased only 2 S and 1 as resting inhibition which would confuse the readers about the importance of encoding
4. Ln 165 - X element of R m*n and Xij element of 0,1 . The variables R m,n I, j not defined because of this the notation also tends to be confusing
5. Ln 179- Figure 2 does not show case the spiking neuron model instead
6. Pg4,5 ,Eqn1-3; the variables and notations are not clearly explained and the explanations given are far below the equations which need to be placed near the eqns
7. Eqn 4 is incomplete ‘otherwise or x<0’ is missing
8. Ln207-212- repeated from the introduction part
9. Ln214- 228- Many of the notations are repeated which can be avoided like BN, Conv etc.. Repeated statements like “Figure.3(b) is our improved spiking ResNet residual block,” Figure 3 is clear but explanations of the residual block architecture seem very confusing
10. Ln241- Eqn 7 is referred but not sure of when equation 7,and when 8 would be used. also eqn8 is not called anywhere else
11. Ln 286 In the SECA module, the channel attention is specified across a single layer and that all channels within the network utilize the same set of parameters during the learning process Or is this channel attention only on the convolution layers and not the other layers.
12. In the Fig 4. K =5 and in the explanation k=3 what do you mean by K if k denotes kernel.
13. How as the k value determined to be 3 and was considered optimal?
14. Ln 361 - Data not provided. Provide this as supplementary material
15. Ln371- What was the rationale used for dividing the dataset as training and testing
16. Instead of using the terms above and below figure, it would be better to consider having a and b part for the figure
17. Ln472,506- The references ‘Kheradpisheh, S. R. and Masquelier’ and ‘Tavanaei, A., Ghodrati, M.,etal’, are repeated

Cite this review as

---

## Round 0.2 · accepted · Accept

Thank you for fulfilling the revision requests. I believe the manuscript is ready for publication.